# Differential Cellular Sensing of Fusion from within and Fusion from without during Virus Infection

**DOI:** 10.3390/v15020301

**Published:** 2023-01-21

**Authors:** David N. Hare, Tetyana Murdza, Susan Collins, Katharina Schulz, Subhendu Mukherjee, Roberto de Antueno, Luke Janssen, Roy Duncan, Karen L. Mossman

**Affiliations:** 1Department of Biochemistry and Biomedical Sciences, Institute for Infectious Disease Research, McMaster University, Hamilton, ON L8S 4K1, Canada; 2Department of Medicine, McMaster University, Hamilton, ON L8S 4K1, Canada; 3Firestone Institute for Respiratory Health, St Joseph’s Healthcare Hamilton, Hamilton, ON L8N 4A6, Canada; 4Department of Microbiology and Immunology, Dalhousie University, Halifax, NS B3H 4R2, Canada

**Keywords:** virus–host interactions, innate signaling, membrane fusion, interferon regulatory factor 3, interferon, virus entry

## Abstract

The physical entry of virus particles into cells triggers an innate immune response that is dependent on both calcium and nucleic acid sensors, with particles containing RNA or DNA genomes detected by RNA or DNA sensors, respectively. While membrane fusion in the absence of viral nucleic acid causes an innate immune response that is dependent on calcium, the involvement of nucleic acid sensors is poorly understood. Here, we used lipoplexes containing purified reovirus p14 fusion protein as a model of exogenous or fusion from without and a cell line expressing inducible p14 protein as a model of endogenous or fusion from within to examine cellular membrane fusion sensing events. We show that the cellular response to membrane fusion in both models is dependent on calcium, IRF3 and IFN. The method of sensing fusion, however, differs between fusion from without and fusion from within. Exogenous p14 lipoplexes are detected by RIG-I-like RNA sensors, whereas fusion by endogenous p14 requires both RIG-I and STING to trigger an IFN response. The source of nucleic acid that is sensed appears to be cellular in origin. Future studies will investigate the source of endogenous nucleic acids recognized following membrane fusion events.

## 1. Introduction

Type 1 interferon (IFN-I) is an important early form of innate defense against viral infection. Impaired IFN-I signaling is associated with sensitivity to otherwise innocuous virus infections [1]. Virus-mediated IFN upregulation results from the recognition of pathogen-associated molecular patterns, predominantly viral nucleic acid. Cells must effectively discriminate viral nucleic acid from self-nucleic acid to quickly respond to infection while avoiding aberrant inflammation [2]. To discriminate between self and viral nucleic acid, cellular pattern recognition receptors (PRRs) recognize and respond to virus-associated motifs or mis-localized nucleic acid. The RIG-I-like receptors (RLRs) RIG-I and MDA5 recognize uncapped 5′ tri-phosphorylated RNA, long stretches of dsRNA and perhaps other elements of the RNA secondary structure [3,4,5,6]. Additional PRRs sense mislocalized DNA or DNA containing bacterial CpG motifs [7]. While cells can recognize viral glycoproteins, these pathways often lead to the upregulation of other inflammatory cytokines than IFN [8].

Cells can also recognize virus-associated physiological stress through a wide variety of pathways [9,10,11]. Stress-sensing pathways can sensitize or enhance the IFN response to infection or even upregulate IFN in the absence of infection. Some stress-sensing pathways modulate IFN signaling components, while others upregulate IFN through the production or release of stimulatory nucleic acid [12]. Membrane perturbation during enveloped virus particle entry at the cell surface or within endosomes is one form of virus-associated physiological stress that contributes to IFN-I upregulation [13,14,15,16].

We previously used purified reovirus-derived fusion-associated small transmembrane (FAST)-p14 protein to model virus membrane fusion in the absence of exogenous nucleic acid [13]. FAST-p14 protein is used by non-enveloped reptilian reovirus to spread via cell–cell fusion. Treatment with complexes containing lipofectamine and purified p14 (p14 lipoplexes) triggers an antiviral response dependent on p14-mediated fusion. While we do not fully understand how membrane fusion leads to antiviral signaling, we found that Ca^2+^ oscillations triggered by enveloped virus particle entry or p14-mediated fusion were important for IRF3 activation and production of IFN-I [14]. The response to inactivated virus particle entry was partially dependent on calcium signaling, with RNA and DNA sensors required for the detection of virus particles containing RNA and DNA genomes, respectively. Alternatively, the response to p14 lipoplexes was fully dependent on calcium. A similar study found virus-like particles, which lack viral nucleic acid, or treatment with fusogenic liposomes were sufficient to upregulate IFN-I through the adaptor protein STING [16]. This STING-dependent response occurs in response to both DNA and RNA viruses and is independent of the DNA sensor cGAS [15].

To understand how membrane fusion in the absence of exogenous nucleic acid converges on adaptors involved in prototypic nucleic acid recognition and signaling, we employed various models of membrane fusion and tested the relative importance of different signaling proteins. In this study, we used both exogenous p14-lipoplexes and endogenous p14 expressed from within the cell under the control of an inducible promoter. The delivery of exogenous p14 is a simplified model of virus–cell fusion, whereas endogenous expression of p14 mimics cell–cell fusion during infection with viruses such as fusogenic reoviruses. We found that fusion-mediated calcium signaling is necessary but not sufficient to induce an antiviral state. Instead, the antiviral response appears to involve the sensing of endogenous nucleic acids. When comparing fusion from exogenous p14 lipoplexes to fusion from endogenously expressed p14, we found that these were sensed differently, with the former requiring predominantly RNA sensors, and the latter requiring both RNA and DNA sensors. These data suggest that the mechanism of sensing virus–cell fusion (fusion from without) differs from that of sensing cell–cell fusion (fusion from within), highlighting the sensitivity and complexity of intrinsic and innate cell sensing [10,11].

## 2. Materials and Methods

### 2.1. Materials

Human embryonic lung (HEL) fibroblasts (American Type Culture Collection (ATCC)) and telomerized human fibroblasts (THF) (a kind gift from Victor DeFilippis) were maintained in Dulbecco’s modified Eagle’s medium (DMEM) supplemented with 10% fetal bovine serum. Vesicular stomatitis virus expressing green fluorescent protein (VSV-GFP; Indiana strain) was grown, and its titer determined using Vero cells. The double-stranded RNA (dsRNA) mimetic poly I:C was resuspended in phosphate-buffered saline (PBS) and diluted in serum-free medium to a working concentration of 20 µM. The Ca^2+^ inhibitor 2-aminoethyl diphenylborinate (2-APB) was reconstituted in MeOH and diluted in serum-free medium to a working concentration of 200 µM.

### 2.2. Fusogenic Liposomes and p14 Lipoplexes

Fusogenic liposomes were made as previously described [17]. In brief, 1,2-dioleoyl-*sn*-glycero-3-phosphoethanolamine (DOPE), 1,2-dioleoyl-3-triethylammonium-propane, chloride salt (DOTAP) and 1,2-dioleoyl-*sn*-glycero-3-phosphoethanolamine-N-(lissamine rhodamine B sulfonyl) (ammonium salt) (LR-DOPE) were purchased from Avanti Polar Lipids, Inc (Alabaster, AL, USA). The lipids were reconstituted in a 1:1 methanol and chloroform mixture and mixed in a weight ratio of DOPE/DOTAP/LR-DOPE of 1:1:0.1. The solvent was evaporated using nitrogen gas and stored at −20 °C under argon gas until the lipid mixtures were resuspended in 20 µM HEPES buffer at a working concentration of 1 mg/mL. The solution was vortexed until resuspended and then sonicated in a bath sonicator for 20 min or until clarity to form small unilamellar vesicles.

p14 lipoplexes were made as described by Noyce et al., 2012. In brief, 4 µg of purified p14 was diluted in 100 µL PBS, and 3 µL of lipofectamine 2000 was diluted in 100 µL of nuclease-free water before combining and incubating for 1 h at room temperature and then diluting to 1 mL in serum-free DMEM.

### 2.3. Generation of Cell Lines

THF RIG-I^−/−^, MDA5^−/−^ and RIG-I^−/−^MDA5^−/−^ were generated by lentivirus-mediated CRISPR-Cas9 KO and puromycin selection as described in Hare et al., 2015. FLAG-RIG-I, FLAG-RIG-I^K270A^ and FLAG-RIG-I^K888/902A^ in pEF-Bos plasmid (a kind gift from Michael Gale) were cloned into pLenti-blast via endonuclease digestion and ligation and used to produce lentivirus particles. THF RIG-I^−/−^MDA5^−/−^ cells were reconstituted with FLAG-RIG-I by transduction and blasticidin selection.

Cell lines expressing p14 under a tetracycline-inducible promoter were generated using a transposon recombination-based PiggyBac vector system [18]. The cDNA sequence of the p14 FAST protein was amplified by PCR using the forward primer GGGGACAAGTTTGTACAAAAAAGCAGGCTTCACCATGGGGAGTGGACCCTCT and the reverse primer GGGGACCACTTTGTACAAGAAAGCTGGGTCCTAAATGGCTGAGACATTATCGATGTTG. A two-step recombination reaction mediated by BP and LR clonase enzymes integrated the p14 gene into the PB-TAG vector.

Three plasmids, i.e., PB-TAG (encoding the gene of interest), pCYL43 (encoding the PBase recombination enzyme) and PB-CAG rtTA (encoding the rtTA protein which initiates gene expression in the presence of doxycycline) were nucleofected into telomerase life-extended human fibroblasts (THF) [19,20]. p14-positive cells were selected for with 3 μg/mL of puromycin. THF-p14 STING^−/−^ and THF-p14 RIG-I/MDA5^−/−^ cells were generated in the same manner, but the plasmids were inserted into the corresponding THF knock-out cell type.

### 2.4. Quantitative RT-PCR

RNA was extracted using TRIzol reagent (Invitrogen, Carlsebad, CA, USA) and treated with DNase (Ambion, Carlsebad, CA, USA) according to the manufacturers’ instructions. Then, 500 ng of RNA was reverse-transcribed using SuperScript II Reverse Transcriptase (Invitrogen, Carlsebad, CA, USA) and random hexamer primers as per the manufacturer’s instructions. Quantitative PCR reactions containing Taqman probes and Universal PCR Master Mix (Applied Biosystems, Carlsebad, CA, USA) with a StepOnePlus Q-PCR instrument (Applied Biosystems) were performed according to the manufacturer’s instructions. The Ct values were calculated, and GAPDH was used as an endogenous control to calculate individual ∆∆Ct values. The ∆∆Ct values of the samples were compared with those of mock-treated samples to calculate the fold change. Taqman probes for human GAPDH (Hs02758991_g1), IFIT1 (Hs03027069_s1), ISG15 (Hs00192713_m1) and CXCL10 (Hs00171042_m1) were used.

### 2.5. Live Cell Fluorescence Microscopy

Ca^2+^ imaging was carried out as described in Hare et al., 2015. In brief, HEL fibroblasts were grown on glass coverslips, pre-loaded with Oregon Green (5 µM) and sulphobromophthalein (100 µM) for 30 min at 37 °C and treated with fusogenic liposomes (1 mg/mL) diluted 1:100 in serum-free DMEM for 1 h. Hanks balanced salt solution (HBSS) was perfused across the cells for 15 min before confocal time-lapse imaging for 10–20 min at 1 frame/second.

### 2.6. Fixed Cell Imaging

Cells were seeded on coverslips and treated with 0.01 μg/mL of doxycycline for up to 48 h. Post treatment, the cells were fixed with formalin for 10 min at room temperature, washed three times in PBS and stained with CellMask Deep Red plasma membrane stain and Hoescht 33,342 1:10,000 in PBS for 10 min. Coverslips were mounted onto slides with ProLong Gold antifade reagent and left to cure overnight at room temperature. Images were acquired on an EVOS FL Auto 2 widefield microscope at 20× magnification.

### 2.7. Plaque Reduction Assay

The plaque reduction assays were carried out as described in Hare et al., 2015.

### 2.8. Western Blotting

Protein extracts were harvested by scraping the cells into modified RIPA buffer (50 mM Tris pH 7.5, 150 mM NaCl, 1% NP-40, 0.5% sodium deoxycholate, 0.05% SDS and 1 mM EDTA) followed by sonication in a bath sonicator and clarification by centrifugation. SDS sample loading buffer was added before boiling the samples for 5 min. Protein was separated by SDS-poly-acrylamide gel electrophoresis (SDS-PAGE) and transferred to low-fluorescence PVDF membranes (LI-COR, Lincoln, NE, USA). The membranes were blocked with Odyssey Blocking Buffer in Tris-buffered saline (TBS) (LI-COR) for 1 h, incubated overnight with anti-RIG-I (Alme-1, Adipogen, San Diego, CA, USA) diluted 1:1000, anti-GAPDH (Santa Cruz) diluted 1:5000, anti-p14 (in-house antibody generously donated by Roy Duncan’s Lab) diluted 1:1000, or anti-β-actin (Sigma-Aldrich A1978) diluted 1:20,000 followed by IRDye 800 conjugated anti-rabbit (LI-COR) or IRDye680 conjugated anti-mouse (LI-COR) diluted 1:5000 for 1 h and scanned on an Odyssey CLx (LI-COR). All antibodies were diluted in 50% blocking buffer in TBS.

### 2.9. Dot Blots

Diluted dsRNA (E200 synthesized as described previously [21]) or purified p14 were applied to a dry Hybond-N+ positively charged nylon membrane (Amersham) and crosslinked with 100 µJ/cm^2^ UV light using a Stratalinker 1800 (Stratagene). The membrane was blocked in 5% skim milk in TBS for 1 h, incubated with J2 antibody (SCICONS) diluted 1:2000 in 1% BSA in TBS for 2 h, followed by HRP-conjugated anti-mouse secondary antibody (Sigma, St. Louis, MO, USA) diluted 1:5000 in 1% skim milk in TBS for 1 h. An ECL solution of luminol, 4-IPBA and H_2_O_2_ in 100 mM Tris, as described [22], was added prior to X-ray film exposures.

## 3. Results

### 3.1. Calcium Is Necessary but Not Sufficient for Antivial State Induction

Given the differential requirement of calcium signaling for p14 lipoplexes versus non-replicating enveloped virus particles, we first tested whether calcium signaling was sufficient to activate an antiviral response using fusogenic liposomes devoid of viral protein or nucleic acid. The liposomes contained fluorescently tagged lipids that bound to and stained primary human embryonic lung (HEL) fibroblasts within 1 h of treatment, indicative of membrane fusion (Figure 1A). To measure Ca^2+^ signaling, we loaded HEL fibroblasts with a Ca^2+^-sensitive dye and used time-lapse fluorescence microscopy. Similar to findings with enveloped virus particles [14], the liposome treatment triggered robust Ca^2+^ oscillations (Figure 1B). In parallel experiments, we performed a standard antiviral plaque reduction assay using vesicular stomatitis virus (VSV) expressing green fluorescent protein [23]. While HEL fibroblasts treated with the dsRNA mimetic poly I:C were protected from the VSV-GFP challenge, fusogenic liposomes failed to trigger an antiviral response (Figure 1C). These data suggest that membrane fusion per se is not sufficient to activate IRF3 and upregulate antiviral ISGs.

### 3.2. RNA Sensors Are Required for the Response to Exogenous p14

We previously showed that p14 lipoplexes trigger Ca^2+^- and IRF3-dependent upregulation of antiviral genes in the absence of exogenous nucleic acid or traditional viral PAMPs [13,14]. Canonical IRF3 activation requires scaffold proteins, such as STING or MAVS, to bring together IRF3 and its kinase TBK1 [24]. To investigate the role of these scaffold proteins in the response to p14-lipoplex fusion, we treated wildtype, STING^−/−^ or MAVS^−/−^ telomerase-immortalized human fibroblasts (THFs) with p14 lipoplexes and evaluated the antiviral response. While the antiviral response was intact in STING-deficient cells, it was absent in MAVS-deficient cells (Figure 2), indicating that components of the RNA-sensing pathway are required for the response to exogenous p14. In the absence of an antiviral response, the addition of lipofectamine or p14-lipoplexes enhanced VSV-GFP infection of THFs, as we previously observed.

### 3.3. The Response to p14 Requires the RNA Binding Sites of RIG-I

While MAVS potentially functions as an adaptor for multiple sensing proteins, the best characterized upstream sensors are RIG-I and MDA5. When we similarly treated wildtype, RIG-I^−/−^, MDA5^−/−^ or RIG-I^−/−^MDA5^−/−^ THFs with p14 lipoplexes, we found that RIG-I^−/−^ THFs had impaired the antiviral (Figure 3A) and ISG (Figure 3B) responses, which were further diminished in RIG-I^−/−^MDA5^−/−^ THFs. These data suggest that RIG-I and MDA5 collectively contribute to the antiviral response to p14 lipoplexes. RNA recognition by RIG-I or MDA5 involves RNA binding to an N-terminal helicase domain and exposure of a C-terminal CARD domain that interacts with MAVS [25]. To investigate the role of the RIG-I RNA binding domains in the recognition of p14 lipoplexes, we reconstituted RIGI^−/−^MDA5^−/−^ THF cells with either wildtype or mutant RIG-I, including the K270A mutant which lacks ATPase activity [26] and the K888/902A mutant which lacks two lysine residues critical for RNA binding [27]. Control experiments confirmed similar expression levels of the wildtype and mutant proteins (Figure 3C). While the reconstitution of RIG-I^−/−^MDA5^−/−^ THF cells with FLAG-RIG-I restored the antiviral response to p14 lipoplexes, reconstitution with either FLAG-RIG-I K270A or FLAG-RIG-I K888/902A failed to do so (Figure 3D,E). This finding suggests that both the ATPase and the RNA binding activities of RIG-I are important for the antiviral response to p14 lipoplexes, suggesting that the response to p14 involves the recognition of dsRNA.

### 3.4. The Response to p14 Is Not Due to Contaminating RNA

While prior studies showing the necessity of the p14 fusion domain suggested contaminating RNA is insufficient to trigger a response [13], these new data prompted additional control experiments. As further evidence against the possibility of dsRNA contamination being responsible for the observed outcomes, we detected no dsRNA in purified p14 using dot blots probed with a dsRNA-specific antibody (Figure 4A). This assay is sensitive to as little as 0.05 ng of purified dsRNA or 0.1 ng of dsRNA combined with p14, whereas an antiviral response requires >1 ng of purified transfected dsRNA (Figure 4B). We also found that Ca^2+^ signaling is essential for the antiviral response to p14 lipoplexes but dispensable for the antiviral response to transfected dsRNA (Figure 4C–E). Thus, while the RNA recognition domains of RIG-I are required for the antiviral response to p14 lipoplexes, the presence of contaminating exogenous RNA is insufficient to explain our results.

### 3.5. Generation of Endogenous p14-Expressing Cell Lines to Model Cell–Cell Fusion

These data collectively suggest that fusion from without, via exogenously delivered p14, triggers an antiviral response with an apparent component of RNA recognition through RIG-I and/or MDA5 that cannot be explained by contaminating exogenous RNA. In the context of a natural reovirus infection, p14 is not involved in virus entry, but rather mediates cell–cell fusion from within. This late fusion event is a characteristic way for syncytial viruses to enhance viral spread. To investigate whether an analogous IFN response was elicited by p14 fusion from within, we generated THF cells with doxycycline (dox)-inducible p14 expression using a Tet-On system (THF-p14). The expression of p14 in THF cells resulted in visible cell–cell fusion, yielding large multinucleate syncytial cells within 16 h of doxycycline treatment (Figure 5A; refer also to Figure 6B). A significant and sustained upregulation of the ISGs IFIT1 and CXCL10 was observed by 40 h after doxycycline treatment (Figure 5B). CXCL10 upregulation appeared to occur in two phases, with a low-level response at 16 h, shortly after cell–cell fusion, followed by a stronger sustained response around 40 h after doxycycline treatment (Figure 5B). Similar to fusion from without, the antiviral response to p14-mediated fusion from within was calcium-dependent, as indicated by the loss of ISG expression in p14-expressing cells treated with the calcium inhibitor 2-APB (Figure 5C). The treatment with 2-APB did not impair cell–cell fusion.

### 3.6. The ISG Response to Endogenous p14 Requires Both STING and RIG-I

As p14-mediated fusion from without and fusion from within both require calcium signaling, we next investigated whether the two modes of fusion were sensed similarly. RIG-I^−/−^MDA5^−/−^ and STING^−/−^ THF cells were modified to express p14 under the control of a Tet-On promoter. Upon doxycycline treatment, these modified knock-out cells expressed similar levels of p14 and displayed a similar extent of cell fusion as THF-p14 cells (Figure 6A–C). THF-p14 STING^−/−^ cells showed no ISG upregulation following cell fusion, while THF-p14 RIG-I^−/−^MDA5^−/− cells^ displayed no IFIT1 and ISG15 upregulation and an impaired CXCL10 response (Figure 6D).

Collectively, these data suggest that p14-mediated fusion from without requires RLRs, while fusion from within requires both RLRs and STING. We have previously shown that the response to membrane fusion from exogenously delivered p14 requires IRF3 [13], similar to the response to non-replicating enveloped virus particles [14,28]. Due to the observed differences in p14 detection, we sought to compare the downstream signaling requirements for fusion of exogenous vs. endogenously expressed p14. The contribution of IRF3 to the antiviral response to endogenous p14 was determined in IRF3^−/−^ THF cells. As was previously observed for transfected p14, the antiviral response to endogenous p14 required IRF3 (Figure 7A,B). Similarly, the response to both exogenous (Figure 7C,D) and endogenous (Figure 7A,B) p14 fusion was dependent on IFN, as the response was lost in both cells treated with the IFN-blocking antibody B18R (Figure 7C) and cells lacking the interferon receptor (Figure 7D).

## 4. Discussion

We previously showed that disruption of cellular homeostasis, such as membrane perturbation, is sufficient to trigger an IRF3-mediated antiviral response in the absence of virus replication, viral nucleic acid or detectable IFN-I production [10,13,14]. Our previous findings suggest that membrane perturbation mediated by enveloped virus particle entry or p14 lipoplex treatment triggers calcium signaling, leading to an IRF3-dependent response [13,14,28]. However, it was not clear from these studies if plasma membrane perturbation and calcium signaling are necessary and sufficient to induce an antiviral response. The absence of an antiviral response to fusogenic liposomes suggests that plasma membrane fusion and its associated calcium signaling per se are insufficient to activate IRF3 and trigger innate immune signaling. This observation is consistent with our data from UV-inactivated virus [14] and p14 lipoplexes (this study), which suggest the involvement of nucleic acid sensing. While Holm et al. observed a STING-dependent, Ca^2+^-independent IFN response to fusogenic liposomes [16], several differences may explain this discrepancy, including cell type differences (murine immune cells versus human fibroblasts) or the methodology used to generate fusogenic liposomes (extrusion versus sonication). Our results with p14 suggest that an element of the response may be related to cell stress and endogenous nucleic acid, the levels of which may differ substantially between experimental designs.

While Ca^2+^ signaling is clearly associated with membrane perturbation, it is not still clear how this contributes to antiviral signaling. Ca^2+^ signaling plays an important role in the activation of STING and may also play a role in the activation of MAVS [29,30,31,32,33]. Ca^2+^ signaling also plays an important role in many intracellular membrane fusion events [34]. Indeed, Ca2+ has been implicated in syncytia formation with reovirus FAST proteins, but was not required for pore formation [35]. In our inducible p14 system, chelating calcium blocked antiviral signaling, but not syncytia formation; in human fibroblasts, p14 lipoplex treatment failed to induce syncytia, yet triggered antiviral signaling. These observations collectively suggest that syncytia formation per se is also not required for antiviral signaling. Further studies employing more specific forms of Ca^2+^ inhibition and more refined methods of Ca^2+^ detection will be required to elucidate the precise role that the observed cytosolic Ca^2+^ oscillations play in innate antiviral immunity.

While p14 lipoplex treatment stimulated an ISG response in an RLR-dependent manner, inducible p14-mediated fusion induced ISGs in an RLR- and STING-dependent manner. External p14 lipoplexes did not accumulate at the cell membrane, but were instead taken up by endocytosis and did not routinely induce syncytia formation. Alternatively, endogenously expressed p14 accumulated at the cell membrane and promoted extensive cell–cell fusion, drastically altering the physiological state of the cell. Thus, differences in p14 activity and degree of stress imparted on the cell may account for enhanced signaling stimulation affiliated with fusion from within. Consistent with this hypothesis, cell fusion by bacteria or PEG has been shown to result in abortive cell division and DNA damage which is sensed by the cGAS–STING pathway [36].

Although the response to p14-mediated fusion does not rely on contaminating exogenous nucleic acid, it remains unknown how these fusion events trigger an endogenous nucleic acid response. As some stress-sensing pathways can trigger the recognition of endogenous RNA [12], one possibility is that p14 fuses internal membranes, releasing compartmentalized nucleic acid and exposing it to cytosolic nucleic acid sensors. However, lipoplexes containing the G2A p14 mutant that creates flickering pores between membranes but is fusion-defective [37] retained the ability to induce innate antiviral signaling (data not shown), suggesting that the fusion of internal membranes is not necessary. Future studies with the G2A mutant may shed additional light on how “membrane perturbation” triggers PRRs implicated in nucleic acid recognition.

The disruption of cellular homeostasis can cause the recognition of endogenous nucleic acid by altering the amount, location or motifs of cellular nucleic acid [2,12]. Viral infection, DNA damage or metabolic stress can release stimulatory RNA or DNA from the nucleus or mitochondria [38,39,40,41]. Oxidative stress or the unfolded protein response to ER stress can cause the accumulation of stimulatory RNA or DNA [42,43]. Typically, endogenous nucleic acid sensing is tempered by RNA-editing enzymes such as ADAR1 that prevent duplex formation [44] and nucleases such as TREX1 and SKIV2L that prevent the accumulation of nucleic acid in the cytosol [45,46]. However, the stress responses can alter the activity of these enzymes. For instance, oxidative stress results in the incorporation of oxidative adducts, commonly, 8-hydroxyguanosine (8-OHG), which stabilizes DNA against TREX1 exonuclease degradation, thereby enhancing its potential to serve as a ligand for innate immune DNA sensors such as cGAS [2].

Multinucleation from cell–cell fusion puts stress on the cell division machinery, as most cells are not equipped to mediate the simultaneous division of multiple nuclei. Syncytial cells have been shown to undergo abortive mitosis, which often results in chromosome fragmentation and the formation of micronuclei [36]. Two recent reports showed bacterial induced cell–cell fusion triggered the cGAS–STING pathway via micronuclei formation, and sensing of cytoplasmic chromatin by cGAS activated innate immune responses in SARS-CoV-2-induced syncytia [36,47]. The analogous findings across all these different cell–cell fusion systems suggests that DNA damage-induced IFN responses may be a common feature of all syncytial cells, regardless of the type of fusogen involved. Conversely, our results highlight an additional role for RNA-sensing pathways, suggesting that there may be multiple RNA- and DNA-sensing pathways at play in response to membrane fusion events with potential overlap. RNA polymerase III has been shown to transcribe AT-rich double-stranded DNA into RNA that can serve as a ligand for RIG-I activation, thereby creating a crosstalk between the RIG-I and STING sensing pathways [48,49]. Further investigation is required to identify the sources of endogenous nucleic acids in the context of membrane perturbation-induced stress responses. Determining which transcripts are recognized following p14-mediated fusion from within and fusion from without may give clues about how different events inherent to viral life cycles trigger innate antiviral responses.

## Figures and Tables

**Figure 1 viruses-15-00301-f001:**
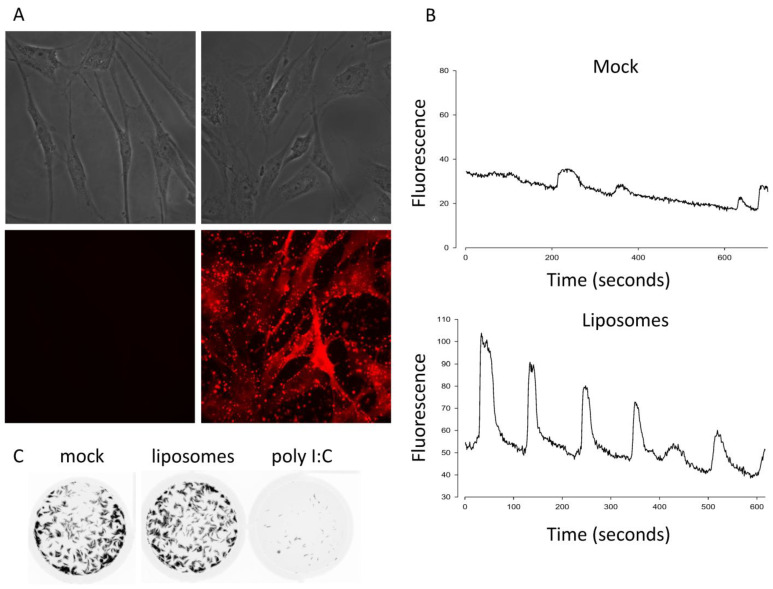
Ca2+ signaling is necessary but not sufficient to induce an antiviral state. (**A**) HEL fibroblasts were treated with fusogenic liposomes containing lissamine rhodamine-conjugated lipids or mock-treated and fixed 1 h later before imaging liposome incorporation by light (top) and fluorescence (bottom) microscopy. (**B**) HEL fibroblasts pre-loaded with a Ca^2+^-sensitive dye were treated with fusogenic liposomes, and the subsequent Ca^2+^ oscillations were imaged by live-cell fluorescence microscopy. Fluorescence from 24 cells per treatment group over 3 biological replicates was plotted over a time period beginning 45 min after treatment, and representative plots are shown. (**C**) HEL fibroblasts were treated with poly I:C or fusogenic liposomes and challenged with VSV-GFP in a plaque reduction assay 16 h post treatment.

**Figure 2 viruses-15-00301-f002:**
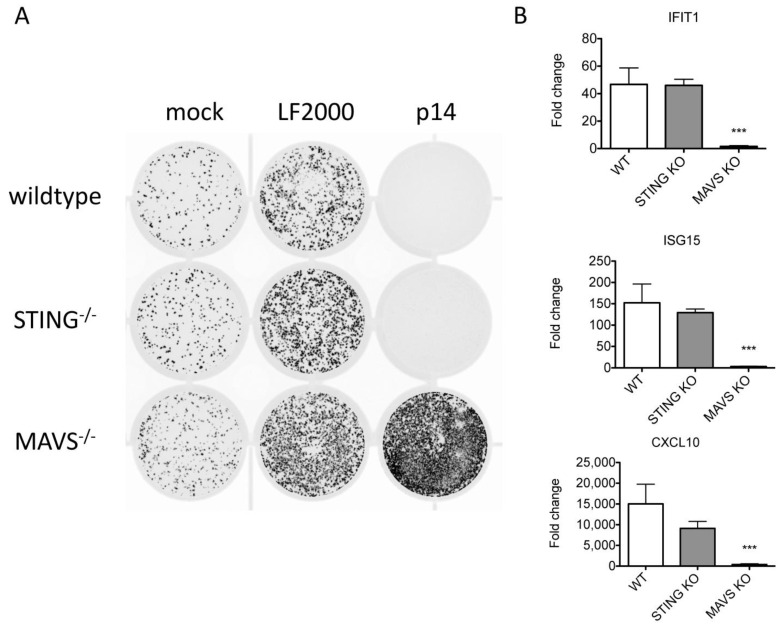
The antiviral response to p14 lipoplexes depends on MAVS but not STING. Wildtype (WT), STING^−/−^, or MAVS^−/−^THF cells were treated with p14 lipoplexes, Lipofectamine 2000 or media alone and (**A**) challenged with VSV-GFP in a plaque reduction assay 16 h later or (**B**) cellular RNA was harvested 12 h later for analysis. Levels of IFIT1, ISG15 and CXCL10 transcripts were measured using quantitative RT-PCR and shown relative to mock-treated THF cells. Group means were log-transformed and compared to that of wildtype (WT) by one-way ANOVA with Bonferroni post-tests (*** for *p* < 0.001).

**Figure 3 viruses-15-00301-f003:**
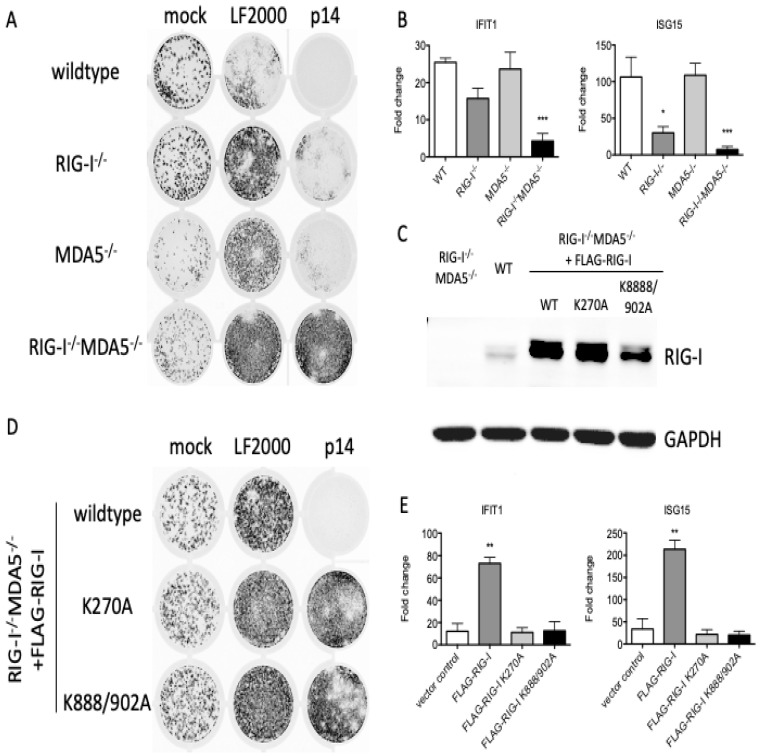
RIG-I RNA binding mutants fail to reconstitute the antiviral response to p14 lipoplexes. (**A**) Wildtype (WT), RIG-I^−/−^, MDA5−/− or RIG-I−/−/MDA5^−/−^THF or (**D**) RIG-I−/−/MDA5−/− THF cells stably reconstituted with FLAG-RIG-I, K270A or K888/902A were treated with p14 lipoplexes or mock-treated for 16 h and then challenged with VSV-GFP for 24 h, and the GFP signal was quantified on a Typhoon imager. (**C**) Protein was harvested from THF cells, and the levels of RIG-I and GAPDH were measured by western blot. (**B**,**E**) In parallel, cellular RNA was harvested 12 h post treatment with p14 lipoplexes, and the levels of IFIT1 and ISG15 transcripts were measured using quantitative RT-PCR and are shown relative to those in mock-treated THF cells. Group means were log-transformed and compared to wildtype (WT) mean by one-way ANOVA with Bonferroni post-tests (* for *p* < 0.05, ** for *p* < 0.01, *** for *p* < 0.001).

**Figure 4 viruses-15-00301-f004:**
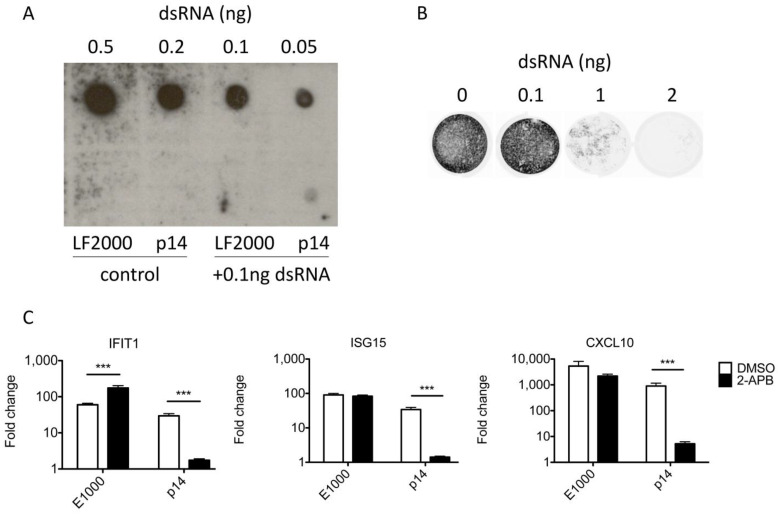
The antiviral response to p14 is not due to contaminating dsRNA. (**A**) Lipofectamine 2000 (LF2000) or 4 µg of purified p14 was incubated for 1 h with or without the addition of 0.1 ng of dsRNA in 5a µL total volume. LF2000, p14 and serially diluted dsRNA were cross-linked to a nylon membrane and probed for the presence of dsRNA using specific antibodies and chemiluminescence. (**B**) THF fibroblasts were transfected with serial dilutions of dsRNA and challenged with VSV-GFP 16 h later in a plaque reduction assay. (**C**–**E**) THF fibroblasts were pretreated with the Ca^2+^ signaling inhibitor 2-APB or DMSO alone, transfected with dsRNA or treated with p14 lipoplexes, and the RNA was harvested 12 h post treatment. (**C**) The levels of IFIT1, (**D**) ISG15 and (**E**) CXCL10 transcripts were measured by quantitative RT-PCR and are shown relative to those in mock-treated THF cells. Groups were compared by two-way ANOVA with Bonferroni post-tests (*** for *p* < 0.001).

**Figure 5 viruses-15-00301-f005:**
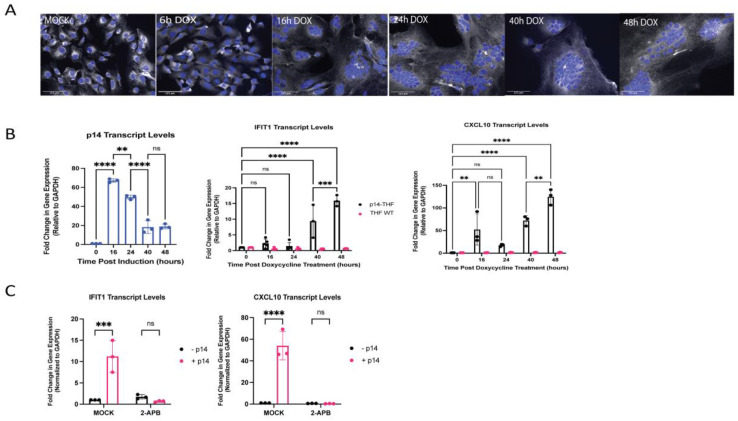
Endogenous expression of p14 triggers cell-cell fusion and an antiviral response in a Ca^2+^-dependent manner. THF-p14 cells were treated with 0.01 µg/mL of doxycycline and monitored over a 48 h time period. (**A**) Cell nuclei were stained with Hoechst 33,342 (blue), and cell membranes were stained with CellMask Deep Red (white). Scale bar indicates a size of 125 µm. (**B**) The presence of p14 in the cells upon doxycycline treatment was verified by RT-qPCR. (**B**) The levels of ISG expression in wildtype THF and p14-THF cells treated with 0.01 µg/mL of doxycycline were assessed by RT-qPCR from RNA harvested at various time points over a 48 h time course. The transcript levels were normalized to GAPDH level and plotted relative to those in uninduced (Time 0 h) cells. Groups were compared by one-way ANOVA (** for *p* < 0.01, *** for *p* < 0.001 and **** for *p* < 0.0001). (**C**) THF-p14 cells were treated with 0.01 µg/mL of doxycycline for 6 h, then treated with 200 µM 2-APB to block calcium release from the ER, and the ISG transcript levels were measured after 48 h.

**Figure 6 viruses-15-00301-f006:**
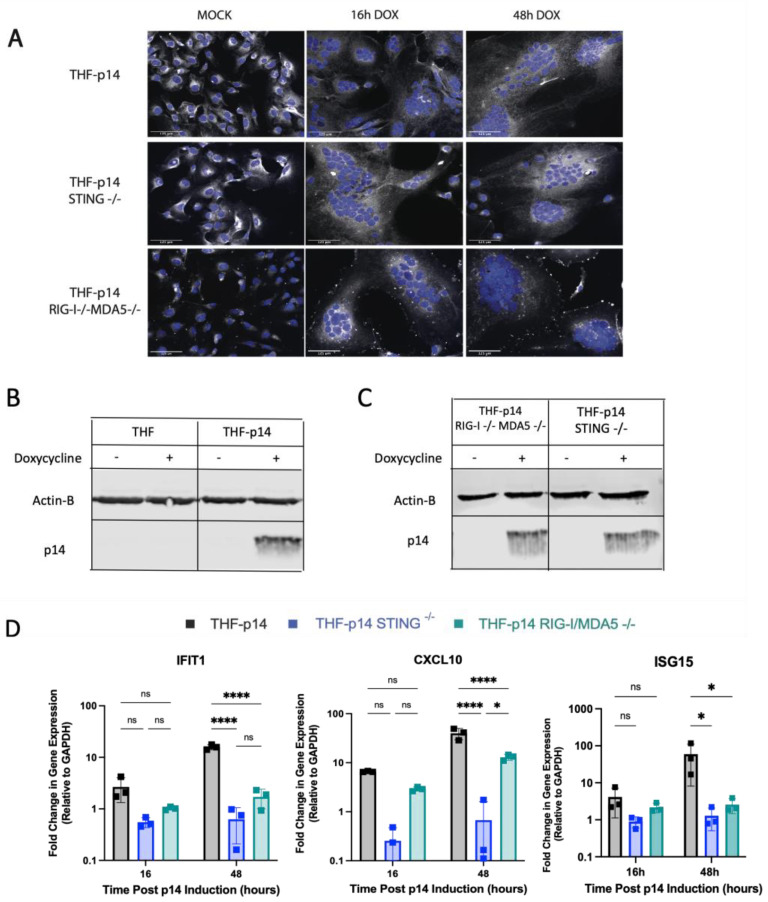
Both STING and RLRs are involved in the ISG response to endogenous p14. (**A**) STING^−/−^ and RIG-I^−/−^ MDA5^−/−^ THF cells were modified to express p14 under a Tet-On inducible promoter. The cell nuclei were stained with Hoechst 33,342 (blue), and the cell membranes were stained with CellMask Deep Red (white). Scale bar indicates a size of 125 µm. (**B**) Protein extracts from p14-THF RIG-I^−/−^MDA5^−/−^, p14-THF STING^−/−^ or p14-THF wildtype cells were separated by SDS-PAGE and probed with p14- and beta actin-specific antibodies. (**C**) Protein extracts from p14-THF RIG-I^−/−^MDA5^−/−^ and p14-THF STING^−/−^ were separated by SDS-PAGE and probed with p14- and beta actin-specific antibodies. (**D**) p14-THF RIG-I^−/−^MDA5^−/−^, p14-THF STING^−/−^ and p14-THF wildtype cells were treated with 0.01 µg/mL of doxycycline for 16 or 48 h, and cellular RNA was harvested for RT-qPCR. The transcript levels were normalized to GAPDH level and are shown relative to those in uninduced cells (Time 0). Groups were compared by two-way ANOVA with Bonferroni post-tests (* for *p* < 0.05 and **** for *p* < 0.0001).

**Figure 7 viruses-15-00301-f007:**
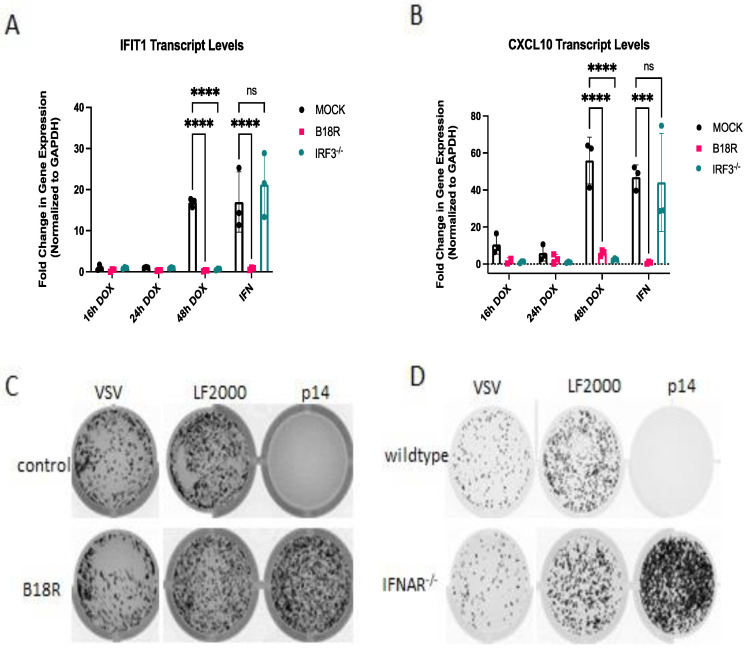
The antiviral responses to endogenously or exogenously delivered p14 requires IRF3 and IFN-I. (**A**,**B**) Wildtype or IRF3−/− THF cells expressing p14 under a doxycycline-inducible promoter were treated with doxycycline for the indicated time and the IFN-blocking antibody B18R where indicated; then, RNA was harvested for qRT-PCR to detect (**A**) IFIT1 or (**B**) CXCL10 transcripts. (**C**) Wildtype THF cells were pretreated with B18R (200ng/mL) to block IFN, or (**D**) wildtype and IFNAR−/− cells were not pretreated and then were transfected with p14/LF2000 and subsequently challenged with VSV-GFP 16 h post treatment. Groups were compared by one-way ANOVA (*** for *p* < 0.001 and **** for *p* < 0.0001).

## Data Availability

The data presented in this study are available on request from the corresponding author.

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
