# Peer review of "Differential Cellular Sensing of Fusion from within and Fusion from without during Virus Infection"

_viruses, 2023, doi:10.3390/v15020301_

Round 1
Reviewer 1 Report
In this manuscript, author explained about the mechanism of sensing virus-cell fusion differs from sensing cell-cell fusion. Author use both exogenous p14-lipoplexes and endogenous p14 expressed from within the cell under control of an inducible promoter.
Exogenous p14 lipoplexes are detected by RIG-I like RNA sensors, whereas fusion by endogenous p14 requires both RIG-I and STING to trigger an IFN response. It is clearly a big effort of author for doing this study, but despite of all the positive outcomes, I have some suggestion where it can be improved.
Comments
1. The title of the paper is little bit confusing. “Differential cellular sensing of fusion from within and fusion from without during virus infection”. It will be good if author can change it.
2. In Figure 5B, the level of CXCL-10 is less at 24 hours post treatment as compared to 16 hours post treatment. Please explain.
3. In Figure 6, please check the labeling A-D. In legends part as well as in explanation it’s not matching with the figure, specially figure 6D.
4. In Figure 6D, Error bar is so high in control at 48 hours post p14 induction during ISG upregulation.
5. In line 335, please give a space between (B) and protein.
6. If author can provide an illustration figure or model of the paper, then it will be good for a paper and as well as it will be more understandable to the readers.
Author Response
We would like to thank this reviewer for his/her thorough reading of the manuscript and thoughtful comments, which we address below in italics:
- The title of the paper is little bit confusing. “Differential cellular sensing of fusion from within and fusion from without during virus infection”. It will be good if author can change it.
We discussed changing the title, but felt the title accurately captures the study and its outcomes. However, to address the reviewer’s suggestion, we have included a graphical abstract to better explain and clarify our study. The addition of this graphical abstract should also serve to answer point 6, below.
- In Figure 5B, the level of CXCL-10 is less at 24 hours post treatment as compared to 16 hours post treatment. Please explain.
It is possible that there are two parts to the CXCL10 response: a low level, early priming response at 16 hours, followed by a second response to the downstream effects of p14. These observations have been added to the results section (lines 326-328).
- In Figure 6, please check the labeling A-D. In legends part as well as in explanation it’s not matching with the figure, specially figure 6D.
Thank you for catching this error, which has been fixed.
- In Figure 6D, Error bar is so high in control at 48 hours post p14 induction during ISG upregulation.
While we see reproducible results with IFIT1 and CXCL10, it is common to see variability with ISG15, which may indicate that induction of this gene is more sensitive to fluctuations within our experimental system. Regardless, all individual data points were significantly higher than the comparators. The data are better reflected when shown on a log scale. As such, we have modified this figure accordingly.
- In line 335, please give a space between (B) and protein.
A space was added.
- If author can provide an illustration figure or model of the paper, then it will be good for a paper and as well as it will be more understandable to the readers.
As indicated in point 1, we have added a graphical abstract to better illustrate the nature and outcomes of our paper. Thank you for this helpful suggestion.
Reviewer 2 Report
In this report, Hare et al., investigated how the membrane fusion starting from outside or within the cell is sensed by using a reovirus p14 fusion protein. They either purified lipoplexes containing this protein or induced expression of this protein by the cell to study fusion from without or fusion from within sensing mechanisms, respectively. Authors found that calcium signaling plays a role in both of these sensing mechanisms, but the detection sensors are different depending on where the fusion is starting from. Authors showed that exogenous p14 mediated fusion is detected by RIG-I sensors whereas endogenous fusion sensing required both RIG-I and STING pathways to stimulate an interferon response. They used human embryonic fibroblasts as wt or generated STING ko, MAVS ko, RIG-I ko, MDA5 ko, or dual ko; RIG-I/MDA-5 ko cell lines to test their hypotheses. They employed VSV infection as a read out for IFN production. They further confirmed the ISG induction with quantitative PCR. Manuscript is well-written, clear and results are supported by adequate experimental data. I have some minor comments and questions:
Authors have not described the calcium sensor used in this study, they just referred to it as Ca sensitive dye, what is it, what concentration did they use?
In Fig 1B, authors showed a Ca oscillation graph, where is the graph for non-treated cells? How many cells were traced (it says multiple cells, how many, why don`t we see multiple waves? What is the x-axis showing?)
In Fig 3C, authors transfected the RIG-I/MDA5 ko fibroblasts with wt, K207A and K888/902A flag RIG-I. They need to show the expression levels of these mutant proteins in comparison to the wt to be able to make a conclusion that they do not sense the fusion and hence do not induce ISGs upon p14 treatment.
In Figure 5C, authors treated the cells with 2-APB to block Ca signaling. Did this also block fusion progressin?
In Figure 7D, why is there such a big difference in plaque reduction assay across IFNAR-/- conditions?
Author Response
We would like to thank this reviewer for his/her careful read of the manuscript and thoughtful suggestions. Our responses are in italics:
Authors have not described the calcium sensor used in this study, they just referred to it as Ca sensitive dye, what is it, what concentration did they use?
This information was added to the materials and methods (lines 150-155).
In Fig 1B, authors showed a Ca oscillation graph, where is the graph for non-treated cells? How many cells were traced (it says multiple cells, how many, why don`t we see multiple waves? What is the x-axis showing?)
We have updated Figure 1 to add mock treated cells, and provided additional information within the figure legend.
In Fig 3C, authors transfected the RIG-I/MDA5 ko fibroblasts with wt, K207A and K888/902A flag RIG-I. They need to show the expression levels of these mutant proteins in comparison to the wt to be able to make a conclusion that they do not sense the fusion and hence do not induce ISGs upon p14 treatment.
We have added the expression levels to Figure 3.
In Figure 5C, authors treated the cells with 2-APB to block Ca signaling. Did this also block fusion progressin?
Fusion is not impaired under these conditions. We have added this information to the text (line 330).
In Figure 7D, why is there such a big difference in plaque reduction assay across IFNAR-/- conditions?
We routinely observe enhanced GFP fluorescence with the use of lipofectamine, which is likely due to lipofectamine enhancing the uptake of VSV. We have added this information on lines 238-240.